# Heat-Labile Toxin from Enterotoxigenic *Escherichia coli* Causes Systemic Impairment in Zebrafish Model

**DOI:** 10.3390/toxins13060419

**Published:** 2021-06-12

**Authors:** Camila Henrique, Maria Alice Pimentel Falcão, Luciana De Araújo Pimenta, Adolfo Luís Almeida Maleski, Carla Lima, Thais Mitsunari, Sandra Coccuzzo Sampaio, Mônica Lopes-Ferreira, Roxane Maria Fontes Piazza

**Affiliations:** 1Laboratório de Bacteriologia, Instituto Butantan, São Paulo 05503-900, SP, Brazil; camila.henrique@butantan.gov.br (C.H.); thais.mitsunari@butantan.gov.br (T.M.); 2Laboratório de Toxinologia Aplicada, Instituto Butantan, São Paulo 05503-900, SP, Brazil; maria.falcao@butantan.gov.br (M.A.P.F.); adolfo.maleski@butantan.gov.br (A.L.A.M.); carla.lima@butantan.gov.br (C.L.); 3Laboratório de Fisiopatologia, Instituto Butantan, São Paulo 05503-900, SP, Brazil; luciana.pimenta@butantan.gov.br (L.D.A.P.); sandra.coccuzzo@butantan.gov.br (S.C.S.)

**Keywords:** heat-labile toxin, Caco-2 cells, zebrafish, systemic effects, cardiotoxic effect

## Abstract

Heat-labile toxin I (LT-I), produced by strains of enterotoxigenic *Escherichia coli* (ETEC), causes profuse watery diarrhea in humans. Different in vitro and in vivo models have already elucidated the mechanism of action of this toxin; however, their use does not always allow for more specific studies on how the LT-I toxin acts in systemic tracts and intestinal cell lines. In the present work, zebrafish (*Danio rerio*) and human intestinal cells (Caco-2) were used as models to study the toxin LT-I. Caco-2 cells were used, in the 62nd passage, at different cell concentrations. LT-I was conjugated to FITC to visualize its transport in cells, as well as microinjected into the caudal vein of zebrafish larvae, in order to investigate its effects on survival, systemic traffic, and morphological formation. The internalization of LT-I was visualized in 3 × 10^4^ Caco-2 cells, being associated with the cell membrane and nucleus. The systemic traffic of LT-I in zebrafish larvae showed its presence in the cardiac cavity, yolk, and regions of the intestine, as demonstrated by cardiac edema (100%), the absence of a swimming bladder (100%), and yolk edema (80%), in addition to growth limitation in the larvae, compared to the control group. There was a reduction in heart rate during the assessment of larval survival kinetics, demonstrating the cardiotoxic effect of LT-I. Thus, in this study, we provide essential new depictions of the features of LT-I.

## 1. Introduction

Bacterial toxins have been increasingly recognized as important virulence factors in a variety of pathogenic bacteria, being responsible for the symptoms associated with several important diseases such as diarrhea [1,2].

Heat-labile toxin I (LT-I) is one of the two toxins produced by enterotoxigenic *Escherichia coli* (ETEC) strains, a pathotype responsible for approximately 10 million cases of diarrhea per year worldwide, mainly affecting children under five years old in low-income areas of developing countries [3,4]. It is also considered the main pathotype that causes “traveler’s diarrhea,” which is associated with visitors in transit through some settings in developing countries—mainly Latin America, the Caribbean (i.e., Haiti and the Dominican Republic), South Asia, the Middle-East, and Africa [3,4,5].

LT-I is an oligomeric bacterial enterotoxin with an AB_5_-type structure, composed of an A sub-unit formed of two domains (A_1_ and A_2_) connected by a disulfide bridge: A_1_ is the enzymatically active portion and A_2_ serves as a support, which is non-covalently linked to the pentameric B sub-unit [6,7,8]. Its internalization occurs mainly through the binding of the B sub-unit to the GM_1_ ganglioside receptor present in the intestinal epithelium, which is responsible for enabling the internalization of the enzymatically active A sub-unit of the toxin [6,7,8].

The active A sub-unit is allosterically activated by ADP ribosylation factors (ARFs) [9,10]. The active A_1_ portion of the LT catalyzes the ADP ribosylation of the heterotrimeric GTPase Gsα, consequently activating adenylate cyclase [11]. This results in an increase in cAMP, leading to the activation of cAMP-dependent protein kinase A (PKA) which, in turn, phosphorylates multiple serine residues within the R-domain of CFTR [12], allowing for ATP hydrolysis and gating of the chloride channel [13].

In addition to their performance in the intestinal lumen, LT-I and its sub-families (LT-IIa, LT-IIb, and LT-IIc) have also been described as potent mucosal and systemic adjuvants [14,15]. Although the cellular and molecular events which promote such immunomodulatory properties have not yet been fully elucidated and described, it has been suggested that the physical interactions generated by the binding of the B pentameric sub-unit with ganglioside receptors located on the surface triggers one or more types of immunocompetent cells [15,16,17].

To observe the action and effects of the LT-I toxin as an adjuvant, different models have been (and/or are) used. The most widespread methodology for observing the effect of the LT-I toxin in vivo is carried out using a loop attached to the intestines of adult rabbits, where a bacterial culture of ETEC or the toxin itself is inoculated into the animals through surgical techniques, generating fluid accumulation in the intestine [18]. Newborn mice have also been used, with the percutaneous injection or oral administration of the bacterial culture supernatant, followed by the removal of the intestine for analysis [19,20,21]. In studies carried out with LT-I and LT-II sub-families as adjuvants, the murine model is the most widespread, in which both enteral and parenteral routes have been employed [14,15,22]. These methodologies are extremely laborious and, at present, are often incompatible with the rules of the committees that regulate ethics in the use of animals.

In this context, the zebrafish (*Danio rerio*) is a vertebrate species with several advantageous characteristics filling the gap between cell cultures and rodents, allowing for early in vivo validation, as well as several human pathogen studies [23,24,25,26,27,28,29], in addition to studies evaluating the neutralization of toxins by antibodies [30], efficiency of molecules as vaccine adjuvants [31,32,33], and vaccine efficacy and safety tests [34]. Zebrafish genes show 70% similarity to those of humans [35], as well as other advantageous characteristics, when compared to models already established in rodents, such as the optical clarity of embryos and larvae, ease of creation and manipulation, high reproductive rate, and rapid development, among others [26,28]. Additionally, zebrafish have a fully functional digestive system that is visible for five days after fertilization, as well as high homology with superior vertebrate organisms, in terms of cell composition [36,37]. Furthermore, the presence of GM_1_ ganglioside receptors has been observed [38]. Considering these advantageous characteristics, combined with the limitations of current animal models used for LT-I toxin studies, the zebrafish presents itself as an interesting model for potential use in studies concerned with LT-I produced by ETEC.

Concerning studies of LT-I in vitro models, adrenal cell lines Y-1 and CHO epithelial cells (Chinese Hamster Ovary) are usually used, enabling several studies, such as those aimed at their detection [39,40], neutralization by antibodies [41], or elucidation with respect to the type II secretion system (T2 SS) [42], among others. In adjuvant studies assessing the mode of action and effect on the immune system by the LT-I toxin, the human cell line derived from colorectal adenocarcinoma (Caco-2) has been widely used [43,44]. This line presents several advantageous characteristics, such as spontaneous differentiation into enterocytes (under normal culture conditions), maintaining the morphological and functional characteristics of the mature cement, and the expression of GM_1_ ganglioside receptors [45,46]. Thus, as a human cell line, Caco-2 cells have characteristics closer to the real target of the toxin LT-I than adrenal and epithelial cells of murine origin do, for use as an in vitro model for the detection and neutralization of the LT-I toxin. Herein, we assessed the analysis of the LT-I toxin pathway employing zebrafish as a substitute for mammal models, as well as human intestinal Caco-2 cells, for validating the in vitro fate of LT-I.

## 2. Results

### 2.1. Toxin and Caco-2 Cell Interaction

Different concentrations of Caco-2 cells were evaluated, and 3 × 10^4^ cells per well was determined as a suitable concentration for all experiments, as higher cell numbers impaired the assay (data not shown).

Rhodamine phalloidin and DAPI fluorophores at 1:200 dilutions allowed for microscopic visualization (Figure 1), where the efficiency of labeling the toxin LT-I with FITC (LT-I/FITC) was confirmed by a fluorescence assay of the interaction of the toxin and Caco-2 cells (Figure 2), and visualizing its internalization in both the cell membrane and the nucleus, thus showing its retrograde transport (Figure 2f).

### 2.2. Systemic Distribution of LT-I Toxin in Zebrafish

The systemic traffic of the LT-I/FITC toxin was monitored in 24 hpf zebrafish larvae for up to 96 hpi. Through fluorescence analysis, internalization of the toxin and its evident presence in the cardiac, yolk, and intestine regions were observed (Figure 3e–h). In contrast to that observed in the larvae, the results obtained with the fluorophore FITC (negative controls, Figure 3a–d) suggest the traffic of LT-I through the intestinal tract and its subsequent elimination.

The traffic of LT-I across the intestinal tract was well-defined in in zebrafish intestine (Figure 4), as, at 24 hpi, the toxin was located in the intestinal bulb; at 48 and 72 hpi, the toxin was present in the mid intestine; and, at 96 hpi, LT-I was encountered in the posterior intestine, which was completely different from the FITC control, in which it was located in the intestinal bulb throughout the analyses.

### 2.3. Phenotypic Assessment in Zebrafish Larvae Exposed to LT-I Toxin

Compared to the control (FITC), 64% of the larvae exposed to LT-I/FITC survived in the 96 hpi (Figure 5a); however, they presented malformation events after 24 hpi (Figure 6; cardiac edema). At 96 hpi, 100% of the larvae showed no swimming bladder, while pericardial and yolk edema also occurred in 100% and 80% of the larvae, respectively, when compared to the control group (Figure 5b). The deaths observed in the control group (20%) within the first 24 h were within the normal range, according to the literature, as were the malformations observed in the spine and tail (which were within the limit of 30%) [47].

The macroscopic morphology visualized under a magnifying lens showed that the LT-I toxin induced significant pericardial and yolk edema in the larvae, compared to the control group. Qualitative measurement of these malformations showed that the LT-I toxin caused yolk edema in the larvae, thus showing significance at 96 hpi, compared to the control group larvae (Figure 7a). Furthermore, it also influenced larval growth, showing a significant decrease, in relation to control larvae, at all analyzed times (Figure 7b).

The cardiac region of all larvae treated with the toxin showed an expressive increase and a statistically significant difference during all periods, compared to the larvae treated only with the FITC control (Figure 7c). The edema in the pericardial area of all larvae treated with the toxin was reflected by cardiac dysfunction, as evidenced by the reduction in heart rate over the analyzed periods, mainly at 96 hpi (Figure 8).

## 3. Discussion

The purpose of the current study was to analyze the LT-I pathway in Caco-2 cells and the feasibility of the use of a zebrafish model in determining the transdermal and systemic fate of this toxin. Significant advances in elucidating the pathway and mechanism of LT in vitro cell lines were employed [40,41,42], with Caco-2 cells having been employed among them. These cells express most transporters, drug-metabolizing enzymes, and normal epithelial receptors, including the ganglioside GM_1_ [45,46]. Its expression in Caco-2 cells is dependent on an increase in the number of cell culture passages [46]. Thus, in this study, we were able to visualize the internalization of the LT-FITC using the same concentration previously employed [41,42]. The analysis was performed by confocal microscopy in the 62nd passage of cells, with a concentration of 3 × 10^4^ cells/well. Orthogonal analysis corroborated the presence of the LT-FITC toxin to be associated with the cell membrane and nucleus. Hence, from these studies, it is clear that toxin translocation from the apical to the basolateral surface of the epithelial cell requires endocytosis and processing in one or more intracellular compartments, as has already been described elsewhere [22,48,49,50].

Animal models have been used to examine the colonization and pathogenesis [18,19,21,51] of disease or immunogenicity and efficacy of ETEC vaccine candidates, including intranasal and orogastric adult mouse models [52,53,54,55], an infant mouse challenge model [56], a rat model [57], a rabbit ileal loop (RIL) model [21], and a reversible intestinal tie adult rabbit diarrhea (RITARD) model [58]. These models do not adequately mimic the full spectrum of the disease observed with human ETEC infection and are further limited, as they are difficult to reproduce, or require surgery and/or death as an endpoint. An ideal model requires the absence of antibiotics or surgical intervention, incorporates an orogastric challenge of bacteria, and demonstrates intestinal colonization, diarrheal disease, and the development of protective immunity.

The zebrafish model emerged to overcome such biases, as it has a fully functional digestive system, which is visible five days after fertilization, as well as high homology with superior vertebrate organisms, in terms of cell composition [36,37]. This has allowed for several studies of gastrointestinal pathologies in larvae and adult fish, such as that observed in studies carried out with *Vibrio cholera* [59,60,61], evidencing the potential of this animal model for intestinal colonization analysis using this type of bacteria. This model, in addition to intestinal colonization, also allows for the specific study of virulence factors produced by diarrheagenic bacteria, having, as an example, the cholera toxin (CT) produced by *Vibrio cholera*—a toxin with which LT-I shares structural elements, affinity for the GM_1_ receptor in eukaryotic cells, and enzyme activity [3,6,7,62]. In a recent study, intoxication of zebrafish embryos by transdermal CT absorption—using the Fish Embryo Acute Toxicity test (FET) [63]—made it possible to assess the role of two proteins in CT retro-translocation, after internalization in the intestinal epithelium [38].

Thus, these and other studies using the zebrafish model were successful, due to their several advantages, in relation to other mammals; mainly consisting of their small size, high fertility, rapid development, and transparency, thus allowing for the capture of images in real-time [26,28]. Although zebrafish can connect the gap between assays based on cell cultures and biological validation in higher vertebrate animals, they provide a far more distant model from humans than other animals, such as rodents; for example, its physiology is not identical to that of humans and several human disorders are difficult to reproduce in this model. In addition, some genes appear as two copies; therefore, it is more difficult to determine functional roles in this species [47,64,65,66,67,68]. As such, the zebrafish model does not replace the use of rodent animals—as these provide data that are more easily extrapolated to humans—but it may serve as an important screening tool for therapeutic biomolecules, drugs, and vaccine candidates, among others, in order to complement tests based on rodents or cells, helping to predict the safety of such biopharmaceuticals and, consequently, reducing the overall costs of biological validation [26,27,28,47,64,67].

In this context, in a trial employing a FET test [63] 24 h after fertilization in zebrafish embryos submitted to transdermal absorption of different concentrations of the LT-I toxin [38], the sensitivity of the embryos was dependent on the increase in concentration of the LT-I toxin. Although the lethality of embryos during survival kinetics analysis was low, evident phenotypes of edema were observed in the cardiac cavity and in the yolk, curvature of the spine, and caudal vein, and loss of pigmentation in the larvae were demonstrated at the highest concentration (see Appendix A).

Given this sensitivity, combined with the transparency advantage of zebrafish larvae, the systemic traffic of the LT-I/FITC toxin was analyzed by microinjection into the caudal vein of zebrafish larvae (see Appendix A). An expressive presence of the LT-I toxin in the cardiac cavity, yolk, and intestine was observed through use of fluorescence microscopy. These results lead us to suggest that the toxin LT-I may have an action on the pericardium of the animals, causing edema and cardiac dysfunction; meanwhile, with its later displacement, it acts on the GM_1_ receptors in the intestine, triggering distension of the yolk region, thus increasing it in volume and compressing the intestinal region. Its presence in these regions corroborates the phenotypes observed due to its toxic action (both in the preliminary FET test and in the microinjection test) of cardiac edema, yolk, and intestinal region. It is noteworthy that the transdermal absorption of LT-I by FET test and the systemic inoculation (through the caudal vein) showed similar phenotypes during the development of the zebrafish embryos and larvae, suggesting the same traffic of the toxin, regardless of its inoculation route.

Moreover, in terms of enterotoxicity, LT-I and CT (commonly known as HTL) have been shown to be involved in the strengthening of immune responses to co-administered antigens, as extensively reviewed by [15,16,69]. While it is evident that HTL are potent adjuvants, their inherent toxicities have precluded their use as adjuvants in human vaccines [16]. This omission is particularly relevant for mucosal vaccines. Mouse models have demonstrated that HTL, when administered by the intranasal route, efficiently binds to the nasal neuroepithelium and is subsequently trafficked, by retrograde transport, along the underlying olfactory nerves to the olfactory bulbs in the brain [70,71]. Such retrograde transport might exert an inflammatory effect on brain tissues. Furthermore, they are highly inflammatory when injected into mice dermis or in subcutaneous tissues. Upon injection, they elicit self-limited local erythema and swelling at the site of injection, which may persist for weeks and eventually resolve without sequelae [72].

In an attempt to produce an HLT in which the potentially serious toxicities were abrogated, recombinant engineering has been employed to produce non-toxic forms of HTL. For this purpose, mutant holotoxins were engineered, which targeted single-point amino acid substitutions at amino acids in the A polypeptides considered to be critical for ribosylation activity [15]. Furthermore, the use of LT-IIb and LT-IIc as adjuvants induced less edema, cellular infiltrates, and general inflammation at the site of intradermal injection [14].

Except from cardiotoxicity after the systemic inoculation of LT-I in zebrafish (see Appendix A), the other effects described in rodents, in spite of the employed routes, the toxicity of LT-I were similar, as established for the first time herein in a zebrafish model. Mortality and clinical signs are usually analyzed, in order to evaluate the innate (non-specific) or adaptive (specific) immune system response. Zebrafish also have a well-maintained adaptive immune system, composed of T- and B-lymphocytes that develop from the thymus and kidneys, respectively [43]. Hence, in the present work, besides serving as a prototype for enterotoxicity, we present *Danio rerio* as an excellent experimental model for studies on the safety of new formulations of HTL as adjuvants, highlighting improvements in both time and cost reduction in research and analyses, as well as the use of Caco-2 cell lines as tools for future efficient analyses concerning the LT-I toxin.

## 4. Materials and Methods

### 4.1. LT Toxin

The LT toxin type I (LTh Lot # 14004) used in this study was obtained from a hybrid toxin, which was purified and lyophilized [43]. It was provided to us by Dr. Elizabeth B. Norton and Dr. John D Clements of the Department of Microbiology and Immunology, Tulane University, New Orleans, Louisiana, United States of America. For the conjugation of the toxin with fluorescein isothiocyanate (FITC; Sigma-Aldrich, Darmstadt, Germany), a concentration of 0.001 g/mL was used; for in vitro and in vivo tests, concentrations of 2.5 and 1.5 μg/mL were used, respectively [38,41].

### 4.2. Caco-2 Cell

Caco-2 cells at the 62nd passage were thawed and cultured in 75 mm^2^ culture flasks (Techno Plastic Products, Switzerland, CH) in DMEM containing 4.5 g/mL glucose (Cultilab, Campinas, SP, Brazil) and 10% fetal bovine serum (FBS; Gibco, Thermo Fischer, Waltham, MA, USA), at 37 °C and with 5% CO_2_, until reaching 70% confluence. To prepare the plates, the cells were removed from the bottle with the addition of 2 mL of trypsin and neutralized with 2 mL of DMEM containing 10% FBS after the observation of cell detachment. The cells were transferred to a 50 mL polypropylene tube for centrifugation at 259× *g* for 5 min at 4 °C. Then, the supernatant was discarded and the precipitate was resuspended in 5 mL of DMEM containing 10% FBS medium. Cell concentrations were adjusted after counting in a Neubauer chamber.

### 4.3. Conjugation of LT with FITC Fluorophore

Conjugation was performed, as previously described [73], using a standardized volume of 0.05 mL of the LT-I toxin at a concentration of 0.001 g/mL, incubated with 0.25 mL of FITC (Sigma-Aldrich, Darmstadt, Germany) and 0.30 mL of HEPES buffer (N-(2-hydroxyethyl) piperazine-N’-(2-ethane sulfonic acid; Sigma-Aldrich, St. Louis, MO, USA) at pH 5.5 for 3 h at a temperature between 2 and −2 °C. The solution was homogenized every 30 min, and 0.60 mL of ammonium chloride solution was added after 3 h. The contents were transferred to a Vivaspin 2 10,000 MWCO column (Sigma-Aldrich, Darmstadt, Germany), and 0.5 mL of HEPES pH 8.3 buffer was added as washing buffer. Then, they were centrifuged at 5.975× *g* for 5 min at 19 °C and five washes were performed with 1 mL of HEPES pH 5.5 buffer, followed by centrifugation at 5.975× *g* for 5 min at 19 °C. The excess of free FITC was removed from the column and stored at −20 °C.

### 4.4. Fluorescence

Caco-2 cells were cultured, detached from the bottle with trypsin, and distributed on 13 mm-diameter sterile glass coverslips contained in 24-well cell culture plates. In order to standardize the cell concentration for better visualization, the concentrations of 1 × 10^3^, 3 × 10^3^, 1 × 10^4^, 3 × 10^4^, and 2 × 10^5^ cells/well were analyzed. For fixation, paraformaldehyde solution (PFA) with Triton X-100 (4% PFA, 5% sucrose diluted in PHEM buffer) was added for 5 min at room temperature. Washing was carried out with pH 6.9 PHEM buffer (2 mM HEPES, 10 mM EGTA, 2 mM MgCl_2_, and 60 mM PIPES) plus glycine (100 mM) for 5 min. Then, PFA solution was added for 15 min, followed by another cycle of washing with PHEM–glycine buffer for 10 min. For the visualization of actin filaments and the nucleus, cells were incubated for 30 min with rhodamine phalloidin (Invitrogen, Thermo Fischer Scientific, Carlsbad, CA, USA) and DAPI (Sigma-Aldrich, Darmstadt, Germany) in dilutions of 1:200 and 1:500, respectively, diluted in PHEM–glycine buffer. The coverslips were coupled to slides added with Vectashield (Vector Laboratories, Burlingame, CA, USA), dried, and subsequently stored at −20 °C. After standardization of cell concentration, the tests were carried out with the addition of LT-I/FITC. For this, the steps mentioned above were performed and, after 18 h of incubation, a volume of 1 mL/well-containing 2.5 μg/mL of the LT-I toxin in DMEM (Cultilab, Campinas, SP, Brazil) containing 2% FBS was added to the cells and incubated at 37 °C, with 5% CO_2_ for 7 h. The fixation and staining steps for the actin filaments and the nucleus, with rhodamine phalloidin and DAPI, respectively, were conducted using the standardized dilution. The coverslips were coupled to the slides added with Vectashield (Vector Laboratories, Burlingame, CA, USA), dried, and subsequently stored at −20 °C. Visualization of the assay was performed using a Confocal Laser Scanning Microscope (Confocal TCS, SP8, Leica, Germany). All experiments were executed independently three times.

### 4.5. Zebrafish Husbandry

The experiments were carried out on the Zebrafish Platform of the Butantan Institute (CeTICs/FAPESP), with adult zebrafish (<18 months of age) of the AB strain (International Zebrafish Resource Center, Eugene, OR) under the following conditions: Room temperature, 28 °C; dark cycle, 14/10 h; and groups separated by sex and raised in individual aquariums present in a specific rack (ALESCO, Monte Mor, SP, Brazil), in order to maintain water throughout the system (60 μg mL^−1^ instantaneous sea salts from the ocean).

### 4.6. Zebrafish Embryo Toxicity Assay (Fet Test OECD 236)

Zebrafish embryo toxicity assays were realized according to the OECD guideline No. 236: Fish Embryo Acute Toxicity (FET) Test. Assays were conducted in 24-well plates (Costar^®^ 24-Well Cell Culture Cluster, Corning Incorporated, NY, USA) with five embryos (4- to 32-cell stage) per treatment/concentration at 28 °C. Embryos were exposed to different concentrations of LT-I toxin diluted in E2 medium, and the experiment was performed in quadruplicate. After treatment, embryos and larvae were monitored with a Leica M205C fluorescence stereomicroscope and/or collected at several time points (24, 48, 72, and 96 hpi), where each individual embryo was scored for mortality (embryo coagulation) and morphological malformations.

### 4.7. Microinjection into Zebrafish Larvae

Zebrafish embryos were collected in spawning traps, and larvae, 24 h post-fertilization (hpf) (final *n* = 43), were transferred to a petri dish (100 × 25 mm) with 5 mL of E2 0.5 × medium (7.5 mM KH_2_PO_4_, 2.5 mM Na_2_HPO_4_, 15 mM NaCl, 0.5 mM KCl, 1 mM MgSO_4_ + 7 H_2_O, 1 mM CaCl_2_ + 2 H_2_O, and 0.7 mM NaHCO_3_) and 0.4% tricaine (ethyl-3-aminobenzoate, #MS-222, Sigma Chemical Co. St. Louis, MO, USA). Under anesthesia, the larvae were placed in another Petri dish with E2 medium and Pronase enzyme (Sigma Chemical Co., St. Louis, MO, USA) at a concentration of 0.02 mg/mL, until the chorions were removed, and observed using a Leica M205 C stereomicroscope (Leica Microsystems, Newmarket Rd, Cambridge, UK). After decorionization, the embryos were washed and kept in E2 medium until the beginning of the experiment.

Larvae with 24 hpf (*n* = 43) were individually injected into the tail vein with 1.5 µg/mL of LT-I/FITC (*n* = 31) or 0.5 µg/mL (*n* = 12) of FITC (as a control) using microcapillaries (# 5242952008 femtotips 930,000,043 with 0.5–0.7 µm, Eppendorf, Hamburg-Nord, Germany) coupled to an Injectman^®^ 4 pneumatic microinjector (Eppendorf, Hamburg-Nord, Germany) pressurized to 2–3 nL (Video S2). The embryos were kept in an oven at 28 °C and analyzed for survival kinetics and malformations after 24 (larvae with 48 hpf), 48 (larvae with 72 hpf), 72 (larvae with 96 hpf), and 96 h (larvae with 120 hpf). The microinjection was performed under a Leica M205 C stereomicroscope and images of each larva (*n* = 43), at each time, were obtained under anesthesia using a Lumar V12 stereomicroscope with Axi-ocam MRC REV 3. Fluorescence was observed using the Axio software Vision^®^ (Carl Zeiss, One North Broadway, NY, USA).

### 4.8. Phenotypic Analysis

During the survival kinetics analysis, 96 hpi larva malformation phenotypes, such as larval growth, malformation of the spine and tail, yolk, and pericardial edema, and the absence of a natural bladder, were observed using a Leica M205 C stereomicroscope.

The analysis of the larval length, circumference of the cardiac region, and yolk, was performed at 24, 48, 72, and 96 hpi in the LT-I/FITC toxin group and the control group (FITC). With the larvae on a glass plate in the lateral position, with the aid of a Leica M205 C stereomicroscope and the Leica Application Suite software (LAS v4.11, Leica Application Suite software, EUA), measurements were performed, emphasizing that the total body length was measured from the head to the tip of the tail. To assess the possible cardiotoxicity of the LT-I toxin, the heartbeats of both groups were recorded at 24, 48, 72, and 96 hpi, under 12.5× magnifications, for 15 s using the ImageJ software (v.1.8.0_172, Leica Application Suite software, EUA).

### 4.9. Zebrafish Euthanasia

The euthanasia of zebrafish larvae was carried out at the end of the experiments, with their immersion in a 4% tricaine solution diluted in 0.5 × E2 medium. Then, the absence of a heartbeat was confirmed using a Leica stereomicroscope (M205 C), after which a 10% bleach solution was added.

### 4.10. Statistical Analysis

All values are expressed as the mean ± SEM. The data were obtained using the analysis of variance (ANOVA) and multiple comparison tests. Differences were considered statistically significant when *p* < 0.01, as determined using the GraphPad Prism software (Graph Pad Software, v6.02, La Jolla, CA, USA, 2013).

## Figures and Tables

**Figure 1 toxins-13-00419-f001:**
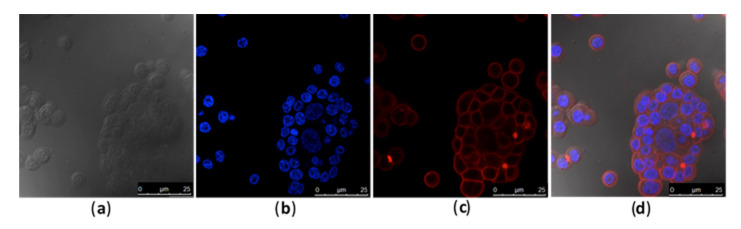
Fluorescence microscopy at the concentration of 3 × 10^4^ Caco-2 cells per well: (**a**) Caco-2 cells, phase contrast; (**b**) Cell nucleus stained with DAPI, in blue; (**c**) Actin of the cell wall stained with rhodamine-phalloidin, in red; (**d**) Overlay of all fluorescence images.

**Figure 2 toxins-13-00419-f002:**
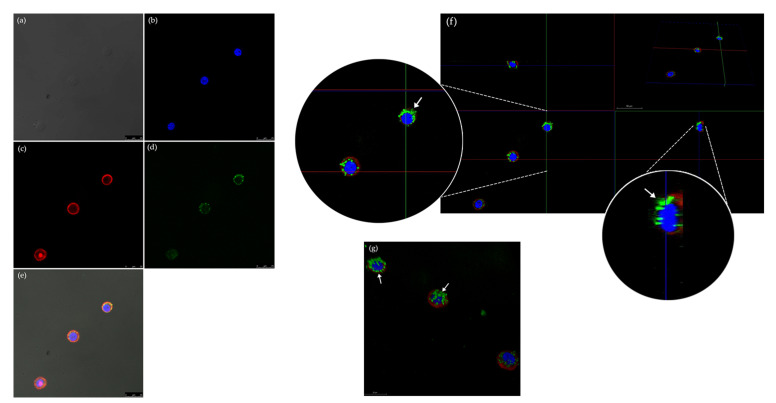
Fluorescence microscopy of LT-I-/FITC interaction with Caco-2 cells for 7 h, visualized with magnification of 63×: (**a**) Caco-2 cells, phase contrast; (**b**) Cell nucleus stained with DAPI, in blue; (**c**) Actin of the cell wall stained with rhodamine phalloidin, in red; (**d**) LT-I/FITC toxin; (**e**) Overlay of all fluorescence images; (**f**) Orthogonal analysis of LT-I/FITC toxin in Caco-2 cells, visualizing the toxin marked with FITC (→) associated with both the cell membrane and the nucleus; (**g**) 3D analysis of LT-I/FITC toxin in Caco-2 cells, evidencing the presence of toxin marked with FITC (→) in the cell membrane and nucleus.

**Figure 3 toxins-13-00419-f003:**
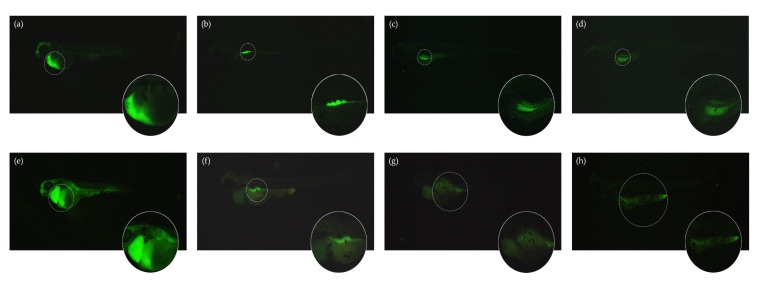
Fluorescence microscopy of systemic traffic of LT-I toxin in zebrafish larvae up to 96 hpi, visualized at 40× magnification: (**a**–**d**) FITC (control) by fluorescence microscopy—(**a**) 24 hpi; (**b**) 48 hpi; (**c**) 72 hpi; (**d**) 96 hpi; (**e**–**h**) LT-I/FITC by fluorescence microscopy—(**e**) 24 hpi; (**f**) 48 hpi; (**g**) 72 hpi; and (**h**) 96 hpi. Visualized using a Lumar V12 stereomicroscope with Axiocam MRC REV 3. Fluorescence deconvolved using the AxioVision^®^ software (Carl Zeiss, Germany), using the following calibration parameters: 0.8× lenses, 52× magnification, −0.10 brightness, 5.78 contrast, and gamma 2.20.

**Figure 4 toxins-13-00419-f004:**
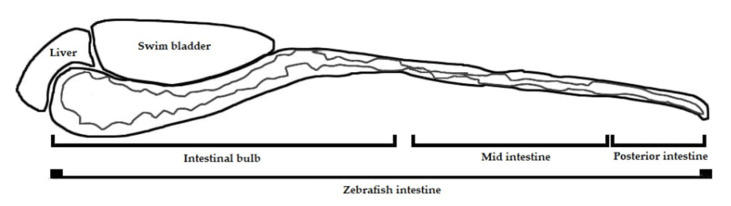
Diagram of zebrafish intestine, showing intestinal bulb, mid intestine, and posterior intestine.

**Figure 5 toxins-13-00419-f005:**
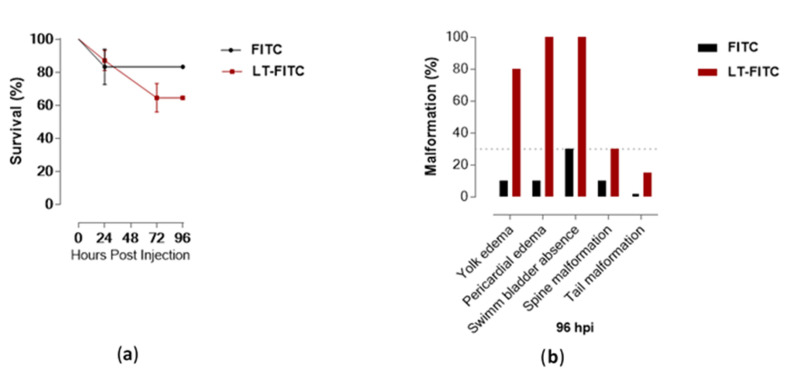
Survival kinetics and malformation events of zebrafish larvae microinjected with the toxin LT-I (*n* = 31) and FITC (*n* = 12) (control): (**a**) Larvae survival kinetics; and (**b**) Malformation events observed at 96 hpi in zebrafish larvae with LT-I toxin.

**Figure 6 toxins-13-00419-f006:**
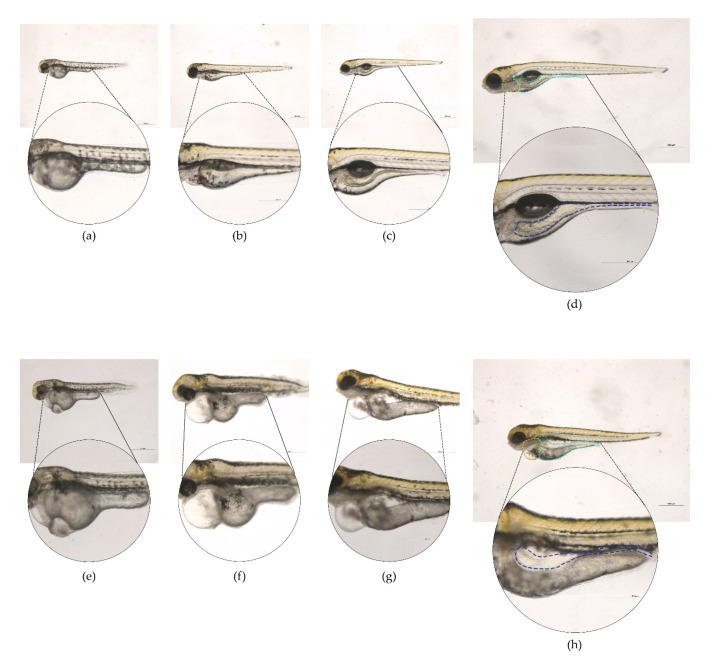
Malformation events observed in larvae injected with the LT-I/FITC toxin and FITC (as control) at different times. Visualized using a Leica M205C LASV 4.11 with software magnification of 28× and 80×: (**a**–**d**) Larvae microinjected with FITC at (**a**) 24 hpi; (**b**) 48 hpi; (**c**) 72 hpi; and (**d**) 96 hpi; and (**e**–**h**) larvae microinjected with LT-I/FITC at (**e**) 24 hpi; (**f**) 48 hpi; (**g**) 72 hpi; and (**h**) 96 hpi. Yellow dotted: heart; light blue dotted: yolk; purple dotted: larva length; and dark blue dotted: intestine.

**Figure 7 toxins-13-00419-f007:**
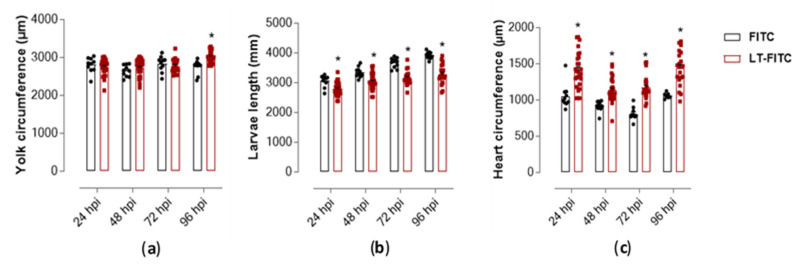
Measurement of heart circumference, yolk, and length of zebrafish larvae injected with toxin LT-I (*n* = 31) and FITC (*n* = 12) (control) at different times of analysis: (**a**) yolk circumference, (**b**) larval length, and (**c**) heart circumference. All values express the mean ± SEM. The differences were considered statistically significant when * *p* < 0.01, as determined using the GraphPad Prism software (GraphPad Software, v6.02, 2013, La Jolla, CA, USA).

**Figure 8 toxins-13-00419-f008:**
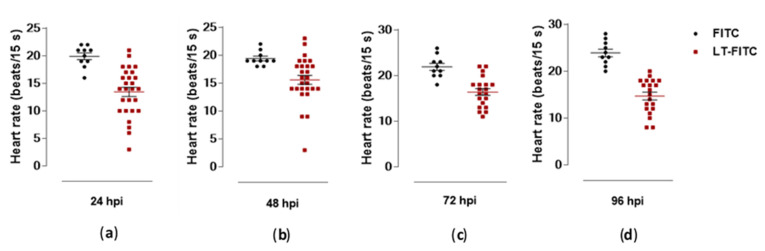
Heart rate at different times of microinjection of zebrafish larvae with toxin LT-I/FITC (*n* = 31) and FITC (control-*n* = 12). (**a**) 24 hpi; (**b**) 48 hpi; (**c**) 72 hpi; (**d**) 96 hpi. All values express the mean ± SEM. The differences were considered statistically significant when *p* < 0.01 as calculated using the GraphPad Prism (Graph Pad Software, v6.02, 2013, La Jolla, CA, USA).

## Data Availability

The data presented in this study are available on request from the corresponding authors. The data are not publicly available due to raw data are not available in the repository of the Butantan Institute (https://repositorio.butantan.gov.br/ (accessed on 14 May 2021)).

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
