# Peer review of "Heat-Labile Toxin from Enterotoxigenic Escherichia coli Causes Systemic Impairment in Zebrafish Model"

_toxins, 2021, doi:10.3390/toxins13060419_

Round 1
Reviewer 1 Report
Review of manuscript: Heat-labile toxin from enterotoxigenic Escherichia coli causes systemic impairment in zebrafish model
The authors have set out to test the effects of LT-I toxin in a zebrafish model because use of other model ”does not always allow more specific studies of how LT-I toxin acts on the systemic tracts and intetinal cell lines”. In general the paper is written in a generalistic and simplistic style. The authors have performed soem potentially interesting experiemtns, but seem unable to put in in context with relevant current literature and convincingly explain the rationale behind the experiments, and what meaningful knowledge they add to our understanding of LT-toxicity. Manuscript seems hastily written with grammatical and spelling errors, missing words, Spanish words and ”,” instead of ”.” in various numbers in the manuscript.
The introduction is short and does not elaborate on what could be the rationale for examining systemic effects of LT, a toxin released by bacteria located in the intestinal lumen, and normally not coming into the blood circulation. The introduction also does not mention what the specific shortcomings other models have that make a zebrafish model neccessary. Regarding CaCo-2 cells the rationale is somewhat better explained, but the large literature of previous studies of LT-I in CaCo cells is hardly mentioned. LT has been studied in CaCo-2 cells with regard to its adjuvanticity, mode of action and effect on immune system for example in Bowman et al. Differential biological and adjuvant activities of cholera toxin and Escherichia coli heat-labile enterotoxin hybrids. Infect Immun. 2001 Mar;69(3):1528-35. PMID: 11179323, and in LOPES et al. Inhibition of T-cell Response by Escherichia coli Heat-Labile Enterotoxin-Treated Epithelial Cells. INFECTION AND IMMUNITY, Dec. 2000, p. 6891–6895 Vol. 68, No. 12.
A better description of the rationale and explanation of what this study adds to our understanding and its correct place in the literature is needed. Some parts of the discussion could be helpful in doing that, and could be moved to the introduction and reformulated.
M&M
From where was the LT toxin obtained?
Otherwise, methods seem well explained. However, I am not very familiar with the zebrafish model.
Results
It is very hard from the images to conclude anything else than that LT has entered the cells. This is hardly a new finding and its novelty needs to be explained and compared to previous literature.
Were the toxin concentrations given in the zebrafish model close to what may be realistic to expect from bacteria infecting animals or humans?
Discussion starts with a mini-review of LT production and function. This would better be rewritten and incorporated as part of the introduction.
There is no discussion of how the findings of internalized LT-FITC adds to other published findings and evidence of retrograde transport.
Starting in line 146 the authors give some more general information on the shortcomings of animal andc ellular models. The statement of ” Surpassing those models zebrafish emerged..” is way too simplistic and optimistic, and should be rephrased into a more realistic view of this models’ potential to add understanding compared to other models.
Line 163 to 174 are repetitive and incoherent.
Lines 176 to 195 convey some potentially interesting systemic effect of LT, but authors need to put this in context of other literature on f.ex the cholera toxin and again if this has any practical meaning for our understanding of ETEC disease or future applications. It is reasonable to claim that good models for testing therapeutic molecules against the Cl secretion effects of LT, but how is the systemic effects on zebrafish model superior for this.
Author Response
Reviewer 1
Comments and Suggestions for Authors
The manuscript was revised taking into consideration your suggestions and other reviewer suggestions. We would like to thank your comments helping us to improve our manuscript. We made an effort to answer the questions, in rearranging the manuscript accordingly. We hope that our answers were satisfactory. All modifications are highlighted in yellow in the text.
Review of manuscript: Heat-labile toxin from enterotoxigenic Escherichia coli causes systemic impairment in zebrafish model
The authors have set out to test the effects of LT-I toxin in a zebrafish model because use of other model ”does not always allow more specific studies of how LT-I toxin acts on the systemic tracts and intetinal cell lines”. In general the paper is written in a generalistic and simplistic style. The authors have performed soem potentially interesting experiemtns, but seem unable to put in in context with relevant current literature and convincingly explain the rationale behind the experiments, and what meaningful knowledge they add to our understanding of LT-toxicity. Manuscript seems hastily written with grammatical and spelling errors, missing words, Spanish words and ”,” instead of ”.” in various numbers in the manuscript.
Answer: in this revised version we put in context our experiments and results, also we carefully checked the spelling and some words specially the legends which we found written in Portuguese, please apologize these misspelling.
The introduction is short and does not elaborate on what could be the rationale for examining systemic effects of LT, a toxin released by bacteria located in the intestinal lumen, and normally not coming into the blood circulation. The introduction also does not mention what the specific shortcomings other models have that make a zebrafish model neccessary. Regarding CaCo-2 cells the rationale is somewhat better explained, but the large literature of previous studies of LT-I in CaCo cells is hardly mentioned. LT has been studied in CaCo-2 cells with regard to its adjuvanticity, mode of action and effect on immune system for example in Bowman et al. Differential biological and adjuvant activities of cholera toxin and Escherichia coli heat-labile enterotoxin hybrids. Infect Immun. 2001 Mar;69(3):1528-35. PMID: 11179323, and in LOPES et al. Inhibition of T-cell Response by Escherichia coli Heat-Labile Enterotoxin-Treated Epithelial Cells. INFECTION AND IMMUNITY, Dec. 2000, p. 6891–6895 Vol. 68, No. 12.
Answer: indeed, we did not include in the first version many important published papers, but in the present version, we rewrite either the introduction or the discussion, most important we added the importance of our model for vaccination studies, since we described the systemic effect of LT-I, many other groups might use zebrafish for their studies.
A better description of the rationale and explanation of what this study adds to our understanding and its correct place in the literature is needed. Some parts of the discussion could be helpful in doing that, and could be moved to the introduction and reformulated.
Answer: Yes, we fully agreed with your comments and therefore we rewrite the manuscript take into considerations your comments and suggestions.
M&M
From where was the LT toxin obtained? Otherwise, methods seem well explained. However, I am not very familiar with the zebrafish model.
Answer: This information in now included in “Materials and Methods” (L. 355-360).LT toxin type I (LTh Lot # 14004) was obtained from hybrid toxin, purified and lyophilized [Bowman and Clements, 2001]. It was provided for us by Dr. Elizabeth B. Norton and Dr. John D Clements, Department of Microbiology and Immunology, Tulane University, New Orleans, Louisiana, United States of America.
Results
It is very hard from the images to conclude anything else than that LT has entered the cells. This is hardly a new finding and its novelty needs to be explained and compared to previous literature.
Were the toxin concentrations given in the zebrafish model close to what may be realistic to expect from bacteria infecting animals or humans?
Answer: The concentrations used for in vitro analysis was 2.5 µg /mL (Ozaki et al., 2015). For parenteral inoculation, usually 0.5 g / mL (Hu et al., 2014). For the tests carried out in FET, we used the concentrations according to the studies by Saslowsky et al., 2010. To evaluate the migration of the toxin, we decided to evaluate a concentration close to that used in in vitro analyzes. All these information’s are now in MM and also mentioned in the discussion
Discussion starts with a mini-review of LT production and function. This would better be rewritten and incorporated as part of the introduction.
Answer: We completely agree, therefore the introduction containing information on the structure and function of the LT-I toxin was reformulated by inserting some information that were before presented in the discussion, as suggested
There is no discussion of how the findings of internalized LT-FITC adds to other published findings and evidence of retrograde transport.
Answer: this discussion is now included in the revised manuscript.
Starting in line 146 the authors give some more general information on the shortcomings of animal andc ellular models. The statement of ” Surpassing those models zebrafish emerged..” is way too simplistic and optimistic, and should be rephrased into a more realistic view of this models’ potential to add understanding compared to other models.
Answer: As suggested, we reformulated the introduction with parts of the discussion. In the introduction, we highlight the animal models and their deficiencies and mention the in vitro models, as well as the advantages related to the use of Caco-2 cells.
Line 163 to 174 are repetitive and incoherent.
Lines 176 to 195 convey some potentially interesting systemic effect of LT, but authors need to put this in context of other literature on f.ex the cholera toxin and again if this has any practical meaning for our understanding of ETEC disease or future applications. It is reasonable to claim that good models for testing therapeutic molecules against the Cl secretion effects of LT, but how is the systemic effects on zebrafish model superior for this.
Answer: As suggested by you, these points are now well discussed and put the use of these models in context
Reviewer 2 Report
The paper titled “Heat-labile toxin from enterotoxigenic Escherichia coli causes 2 systemic impairment in zebrafish model” is an interesting article that provides new and important insights about the vertebrate model of non-mammalian zebrafish (Danio rerio) such as tool for studies LT-I toxin. The study presented the feasibility of using the models in zebrafish and Caco-2 cell lines as tools for future systematic analyses with the LT-I toxin, as well as for new therapeutic molecules screening, to reverse the effect generated by the action of the LT-I toxin produced by ETEC strains.
General comments are that the Authors should read the Instructions for Authors in “Toxins” use the Microsoft Word template or LaTeX template eg. Name, Surname, e-mail after affiliation etc., References. I would like to congratulate the Authors this amazing paper. All the best and stay safe
Minor comments
Line 71 and line 72; FITC (….) something is missing \
Line 77; I suggest “Figure 3a–d”
Line 138; [9,10,11,26] should be changed [9-11,26]
Line 147; [14, 27, 12, 28] should be changed [12,14,27,28]
Line 148; [29,30,31,32] should be [29-32]
Line 159; [38,39,40,41,36] should be [36,38-41]
Line 162; [42,43,44] should be [42-44]
Line 167; [47,48,49,50] should be [47-50] The Authors should check Instruction for the Authors, this same in line 172
Line 322; “References” should be prepared according Instructions for the Author in “Toxins”
Author Response
Reviewer 2
Comments and Suggestions for Authors
The paper titled “Heat-labile toxin from enterotoxigenic Escherichia coli causes 2 systemic impairment in zebrafish model” is an interesting article that provides new and important insights about the vertebrate model of non-mammalian zebrafish (Danio rerio) such as tool for studies LT-I toxin. The study presented the feasibility of using the models in zebrafish and Caco-2 cell lines as tools for future systematic analyses with the LT-I toxin, as well as for new therapeutic molecules screening, to reverse the effect generated by the action of the LT-I toxin produced by ETEC strains.
The manuscript was revised taking into consideration your suggestions and other reviewer suggestions. We would like to thank your comments helping us to improve our manuscript. We made an effort to answer the questions, in rearranging the manuscript accordingly. We hope that our answers were satisfactory. All modifications are highlighted in yellow in the text.
General comments are that the Authors should read the Instructions for Authors in “Toxins” use the Microsoft Word template or LaTeX template eg. Name, Surname, e-mail after affiliation etc., References. I would like to congratulate the Authors this amazing paper. All the best and stay safe
Answer: all the manuscript was revised according instructions for authors.
Minor comments
Line 71 and line 72; FITC (….) something is missing
Minor comments
Line 71 and line 72; FITC (….) something is missing \
Line 77; I suggest “Figure 3a–d”
Line 138; [9,10,11,26] should be changed [9-11,26]
Line 147; [14, 27, 12, 28] should be changed [12,14,27,28]
Line 148; [29,30,31,32] should be [29-32]
Line 159; [38,39,40,41,36] should be [36,38-41]
Line 162; [42,43,44] should be [42-44]
Line 167; [47,48,49,50] should be [47-50] The Authors should check Instruction for the Authors, this same in line 172
Line 322; “References” should be prepared according Instructions for the Author in “Toxins”
Answer: We appreciate the considerations and point out that the comments regarding the mentioned lines have been changed, as well as the revision of the other references, according to the instructions for the author.